# Magnetic-Property Assessment on Dy–Nd–Fe–B Permanent Magnet by Thermodynamic Calculation and Micromagnetic Simulation

**DOI:** 10.3390/ma15217648

**Published:** 2022-10-31

**Authors:** Zhiming Dai, Kai Li, Zhenhua Wang, Wei Liu, Zhidong Zhang

**Affiliations:** 1Shenyang National Laboratory for Materials Science, Institute of Metal Research, Chinese Academy of Sciences, Shenyang 110016, China; 2Jiangsu Key Laboratory of Modern Measurement Technology and Intelligent Systems, Department of Physics, Huaiyin Normal University, Huai’an 223300, China; 3School of Materials Science and Engineering, University of Science and Technology of China, Hefei 230026, China

**Keywords:** micromagnetic simulation, permanent magnet, magnetization reversal, CALPHAD, microstructure

## Abstract

Heavy rare-earth (HRE) elements are important for the preparation of high-coercivity permanent magnets. A further understanding of the thermodynamic properties of HRE phases, and the magnetization reversal mechanism of magnets are still critical issues to obtain magnets that can achieve better performance. In this work, the Nd–Dy–Fe–B multicomponent system is investigated via the calculation of the phase diagram (CALPHAD) method and micromagnetic simulation. The phase composition of magnets with various ratios of Nd and Dy is assessed using critically optimized thermodynamic data. γ-Fe and Nd_2_Fe_17_ phases are suppressed when partial Nd is substituted with Dy (<9.3%), which, in turn, renders the formation of Nd_2_Fe_14_B phase favorable. The influence of the magnetic properties of grain boundaries (GBs) on magnetization reversal is detected by the micromagnetic simulations with the 3D polyhedral grains model. Coercivity was enhanced with both 3 nm nonmagnetic and the hard-magnetic GBs for the pinning effect besides the GBs. Moreover, the nucleation and propagation of reversed domains in core-shell grains are investigated, which suggests that the magnetic structure of grains can also influence the magnetization reversal of magnets. This study provides a theoretical route for a high-efficiency application of the Dy element, realizing a deterministic enhancement of the coercivity in Nd–Fe–B-based magnets.

## 1. Introduction

High-performance Nd–Fe–B-based permanent magnets have an increasing number of requirements in numerous energy-efficient industrial applications, such as wind turbines, pure electric/hybrid vehicles, and automatization [1,2]. Stoichiometry compound Nd_2_Fe_14_B (τ1) shows an extremely high theoretical maximal energy product (~512 kJ/m^3^) and a high anisotropy field μ_0_H_A_ (~7 T) at room temperature, which provides the main part of magnetic properties in this system. However, in real sintered magnets, coercivity is far below the limit that could be expected from a certain fraction of the anisotropy field (*H*_A_) of the τ1 phase [3]. In addition, the poor temperature stability of Nd–Fe–B-based magnets leads to remanence and coercivity decreasing rapidly when the operating temperature rises over 150 °C [4,5]. To compensate for the loss and satisfy the application needs, Nd is usually partially substituted with heavy rare-earth (HRE) elements such as Dy or Tb, resulting in a higher anisotropy field *H*_A_ of the τ1 phase [6].

However, the HRE magnetic moment coupled antiparallel with the Fe moment in Nd_2_Fe_14_B-based compound leads to the decrease of total magnetization. On the other hand, the natural abundance of Dy with respect to Nd is only about 10%, so the mass usage of Dy-substituted Nd–Fe–B magnets is not economical and sustainable, in the long term. Therefore, technological route in enhancing coercivity with the addition of a small amount of HRE is still the focus of research interest in permanent magnet community.

As an extrinsic magnetic property, the coercivity of a magnet is not only associated with the intrinsic magnetic properties of the material, but can also be influenced by the microstructure of the magnet, such as grain size, shape, and boundary, and phase interfaces [7,8,9,10,11,12,13]. Concentrating the HRE elements at the grain surfaces forming a core-shell structure by diffusing the HRE along the grain boundaries into the τ1-phase grains, known as the grain boundary diffusion process (GBDP), is an effective method to increase coercivity [10,14]. The presence of defects or roughness at the τ1-phase surfaces of the sintered magnets is assumed to be the nucleation site for demagnetization that thereby decreases coercivity. The addition of HRE elements during GBDP can repair these surface problems. The compositional design can also be performed by blending precursor powders with the different grain sizes and chemical compositions of Dy alloy powders, resulting in homogeneous HRE distribution [15]. Another available method to design the microstructures in permanent magnets is achieved by growing the materials layer by layer using the vacuum sputtering system [16]. All these processes experience a complex thermodynamic evolution. Since the phase constitutions and morphology of the microstructures intimately depend on the chemical compositions of the magnets and complex heat treatment processes, knowledge of the thermodynamics and phase equilibria among the main and grain boundary phases is indispensable to improve the magnetic properties, and further develop low HRE sintered magnets with considerable performance [17,18].

Here, we theoretically study the phase composition, microstructure, magnetic properties, and coercive mechanisms of the Nd–Dy–Fe–B multicomponent system using the CALPHAD method and micromagnetic simulation. The thermodynamic basement of the Nd–Dy–Fe–B system was optimized with the available experiment data. The analysis of the phase composition and evolution shows that the slight substitution of Dy for Nd is beneficial to the formation of the τ1 phase in both stable and metastable processes. On the basis of the possible phase composition, we designed several 3D simulated models to explore the magnetization reversal behaviors of magnets with different types of GBs and grains. The magnetic properties of GBs, including the angles of adjacent GBs, and the microstructures of grains are important for the nucleation and propagation of the reversed domains. These simulation results provide a theoretical route to efficiently apply Dy to Nd–Fe–B-based magnets.

## 2. Results and Discussion

The thermodynamics and phase equilibria of the Nd–Dy–Fe–B multicomponent system were first assessed by using the CALPHAD method. CALPHAD type thermodynamic databases were developed through a thermodynamic optimization process on the basis of all the available thermodynamic information for compounds and solution phases, and the critically assessed phase diagram data obtained from related studies [19,20,21,22,23,24]. In a thermodynamic database, the Gibbs energies of specific phases are described with different thermodynamic models. Solution phases such as liquid, BCC, FCC, and HCP et al. are described by using the substitutional solution model and expressed as follows [25].
(1)Gθ=∑i=Nd,Dy,Fe,BxiG0iθ+RT∑i=Nd,Dy,Fe,Bxiln(xi)+Gexmθ+Gmagmθ
where *θ*, *R,* and *T* are the solution phases, the gas constant, and the temperature in Kelvin, respectively. ^0^*G_i_^θ^* denotes the molar Gibbs energy of pure element *I* in the *θ* phase taken from the SGTE Unary database [26], which includes the Gibbs energy of both stable and metastable crystal structures. *x_i_* is the mole fraction of element *i* in the *θ* phase. *^ex^**G_m_^θ^* is the excess molar Gibbs energy of the *θ* phase that is involved as a function of temperature and element composition. *^mag^**G_m_^θ^* is the magnetic contribution to the molar Gibbs energy of the magnetic phase, which is given as follows [27]:(2)Gmagmθ=RTln(β+1)g(TTc)
(3)$$g\left(\tau \right)=\begin{array}{c} 1-\frac{1}{M}\left[\frac{79\tau^{-1}}{140p}+\frac{474}{497}\left(\frac{1}{p}-1\right)\left(\frac{\tau^{3}}{6}+\frac{\tau^{9}}{135}+\frac{\tau^{15}}{600}\right)\right]\;\left(\tau \leq 1\right)\\ -\frac{1}{M}\left[\left(\frac{\tau^{-5}}{10}+\frac{\tau^{-15}}{315}+\frac{\tau^{-25}}{1500}\right)\right]\;\left(\tau >1\right)\; \end{array}$$

(4)$$M=\frac{518}{1125}+\frac{11,692}{15,975}(\frac{1}{p}-1)$$where *β* and *T*_c_ are the Bohr magnetons and the Curie temperature, respectively; *τ* is the normalized temperature, which is expressed as *τ* = *T*/*T*_C_. Constant *p* is 0.4 for BCC structures and 0.28 for other phase structures. The intermetallic compounds are treated as stoichiometric compounds, including the binary systems of B–Nd, B–Dy, B–Fe, Dy–Nd, Dy–Fe, and Fe–Nd, and the ternary systems of Nd–Fe–B, Dy–Fe–B, Nd–Dy–Fe, and Nd–Dy–B, which were described using the sublattice model [28]. Among these systems, the thermodynamic basements of B–Nd [29], B–Dy [30], B–Fe [31], Dy–Fe [32], Fe–Nd [33], and Nd–Fe–B [29,34] were reported in several studies with critical thermodynamic data; thus, the thermodynamic optimization process of the Nd–Dy–Fe–B system was launched on the direct basis of these data. The optimized results of the model parameters are listed in Table 1. The comparison of the calculated phase diagram results with the experimental data determined by Gribe and Kobzenko et al. is presented in Appendix A, and proves that the optimized thermodynamic data were reliable. In addition, as the Nd and Dy elements have similar chemistry properties, Nd and Dy may be able to substitute each other with a random ratio in RE_2_Fe_14_B (τ1) and RE_1_._1_Fe_4_B_4_ (τ2) compounds, which are treated as a solid solution.

Using the Gibbs energy functions obtained from the above approach, the thermodynamic calculations containing the Gibbs energy-minimizing routine were conducted under various conditions using the open-source thermodynamic software packages of Open CALPHAD [35]. To understand the influence of Dy substitution for Nd on phase reactions and τ1 phase formation, the calculated vertical sections of Nd_14_Fe_78_._5_B_7_._5_–Dy_14_Fe_78_._5_B_7_._5_, Nd_14_Fe_80_B_6_–Dy_14_Fe_80_B_6_, and Nd_2_Fe_14_B–Dy_2_Fe_14_B under a temperature in the range of 1800–1400 K are shown in Figure 1a. The stable peritectic reactions were γ-Fe + Liq = τ1 and Dy_2_Fe_17_ + Liq = τ1 in both the Nd-rich and the Dy-rich side, which does not favor the formation of τ1 during a stable cooling process. However, a small Dy substitution of Nd narrowed the formation region of γ-Fe and suppressed γ-Fe formation during a metastable process with a certain supercooling temperature. When the amount of Nd and Dy increased to 14% of the total atoms, τ1 became the primary phase forming from the liquid solution under a certain amount of Dy (about 4.6–7.2% for X(Fe) = 80% and about 3.5–9.3% for X(Fe) = 78.5%). On the other hand, the Dy substitution of Nd could also enhance the transformation driving force (TDF) of τ1 from the liquid solution phase or amorphous. Given that the TDF of the RE_2_Fe_17_ phase was close to τ1, a comparison of TDF between τ1 and RE_2_Fe_17_ when the cooling temperature was set below 1200 K is given in Figure 1c. In the absence of Dy, the TDF value of RE_2_Fe_17_ exceeded the TDF of τ1 once the atomic ratio of Nd:B was larger than 2:1, which was also one of the reasons for the slightly larger amount of B always being designed in raw materials in comparison with the ratio of 2:14:1 in the Nd_2_Fe_14_B phase during commercial Nd–Fe–B magnet production.

The high coercivity of the Nd–Fe–B-based magnet requires the main phase grains to be isolated by the nonmagnetic RE-rich grain boundary. Thus, component designs of the magnets consist of a high concentration of τ1 and a slight amount of RE-rich alloys. The concentration distributions of τ1 (N(τ1)) and Nd-rich (N(DHCP)) in the Nd_11+x+y_Dy_2_Fe_80–y_B_7−x_ multicomponent system are shown in Figure 2c,d, respectively. The mole fractions of τ1 and Nd-rich phases were calculated using the optimized thermodynamic basement, and typical results under various temperatures are shown in Figure 2a,b. The increase in the value of x could enhance both of N(τ1) and N(DHCP) before it reached about 1.3, where the stable formation phases changed from τ1, DHCP, and τ2 to τ1, DHCP, and Nd_5_Fe_17_ (as shown in Figure 1b). As a result, the amount of N(τ1) rapidly decreased.

In order to achieve a higher application efficiency of Dy, it is necessary to understand the relationship between each element or phase distribution and coercivity in the Nd–Dy–Fe–B system. The dynamic magnetization reversal processes of multicomponent magnets were investigated with the Object Oriented Micromagnetic Framework (OOMMF) method by solving a Landau–Lifshitz–Gilbert (LLG) equation with various applied fields [36]. The direction of magnetization in each node could be determined by minimizing the total Gibbs energy *E_t_*, which can be expressed as follows [37,38]:(5)$$E_{t}=E_{ex}+E_{k}+E_{d}+E_{H}=\int^{\text{​}}\left[-\mu_{0}M_{s}m\cdot \left(H_{\mathrm{ext}}+\frac{1}{2}H_{d}\right)-K{\left(m\cdot u\right)}^{2}+A{\left(\nabla \cdot m\right)}^{2}\right]d^{3}r\;$$
where *E_ex_*, *E_k_*, *E_d_*, and *E*_H_ are the exchange energy, magnetocrystalline anisotropy energy, demagnetization energy, and Zeeman energy, respectively. ***m***, ***u***, ***H_ext_***, ***H_d_***, *K*, and *A* are unit magnetic moment, unit vector, external field, demagnetic field, anisotropic constant, and exchange constant, respectively.

For the simulated models to be more similar to real magnets, the characteristics of polyhedral grains, an independent orientation for each grain, a core-shell structure, and grain boundaries were modeled as illustrated in Figure 3a. The reference micromagnetic parameters of Nd_2_Fe_14_B, Dy_2_Fe_14_B, α-Fe, and Nd-rich phases are listed in Table 2.

In several previous works, experimental studies were reported on the correlation between the composition of the GB phases and its magnetic behavior [3,39,40]. However, it is still challenging to experimentally design the composition of GBs and magnetic properties without any other changes. Thus, micromagnetic simulations were first employed to explore the influence of the magnetic properties of the GB phases on coercivity, as shown in Figure 4a–h. Four 3D models were produced with different GB conditions, namely, without GB (“wo GB” in Figure 4a), nonmagnetic GB (“nonmag GB” in Figure 4b), soft-magnetic GB (“soft-mag GB” in Figure 4c), and hard-magnetic GB (“hard-mag GB” in Figure 4d). The simulated initial magnetization and demagnetization curves of the four models are represented in Figure 3b. Both the nonmagnetic and hard-magnetic GBs could enhance coercivity compared with the model without GBs. The large saturating field of the model with nonmagnetic GB during the initial magnetization process suggests that the polyhedral grains were exchange-decoupled and resulted in the isolated switching of magnetization. The corresponding visible processes of magnetization reversals are presented in Figure 4e–h. The magnet without GBs preferentially nucleated at the grains, whose c axis orientation deviated from the external field direction (z axis) with a large angle. With a larger applied negative external field, the propagation of the reversed domains expanded from the nucleation site into the neighboring grains due to the intergranular exchange coupling. Then, the waterlike reversed domains spilled from the edges of the simulated model and covered the whole space. The magnetization reversals were dominated by both the external and the demagnetization fields. In sintered magnets, the width of a GB is ~2–5 nm, which is comparable to the exchange reference length ($L_{\mathrm{ex}}={\left(A/K\right)}^{1/2}=$1.35 nm) of Nd_2_Fe_14_B [41]. In this work, the width of the GBs was about 3 nm. Therefore, when the nonmagnetic GBs were considered in the system (Figure 4f), the exchange coupling was suppressed; thereby, the nucleation and magnetization reversal behaviors of each grain were isolated as in a single domain state. This kind of GB can always be obtained when the formation phases contain a few Nd-rich phases, such as DHCP_Nd, NdO, and Nd_2_O_3_. However, some experiment measurements revealed that a substantial amount of Fe and Co also existed in the GBs, which may have resulted in the formation of soft-magnetic GBs [40]. The soft-magnetic GB phases preferentially switch at a low external field (Figure 4g), and the directly exchange coupling between grains and GB phases led to a substantial reduction in coercivity. The magnetization reversal occurred from several nucleation sites, and the propagation of reversed domains spread rapidly along the GBs to the whole magnet. Another type of GB was designed into the hard-magnetic Dy_2_Fe_14_B phase (Figure 4h). The direct exchange coupling between the hard-magnetic GB and grain phases could also induce the magnetization reversals of the hard-magnetic GB phase. This is different from the soft-magnetic GBs, as hard-magnetic GBs in a system act as pinning sites against the propagation of reversed domains. The consistent orientation of saturating magnetization in hard-magnetic GBs induced a low permeability, resulting in a similar magnetization reversal behavior with that of nonmagnetic GBs.

Additionally, we explored the correlation between the included angle of adjacent GBs and the magnetization reversal with simple 2D models, as shown in Figure 5a–i. As the initial nucleation sites were set at the vertex of the angle, the propagation processes of the reversed domains presented a significant difference around the GBs with different magnetic properties. A small included angle of the soft-magnetic GBs was beneficial to the reversed domain expansion (as seen in Figure 5a–c). However, when the GBs were substituted by the nonmagnetic or hard-magnetic type, the expansion of the reversed domains was suppressed by the GB pinning effect, and the propagations of the reversed domains expanded preferentially along the direction with large GB included angles. Given that the GB included angle is determined by the shape of the grain in real magnets, designing appropriately shaped main phase grains on the basis of the magnetic properties of GBs is also important to manipulate the magnetization reversal in Nd–Fe–B-based magnets.

In sintered magnets, RE element distributions in main phase grains are always expected to be uniform after several heating processes. However, unexpected defects can be created at grain surfaces during mechanical milling, crushing, or oxidation. The grain boundary diffusion is an effective technology to revise these defects, and the inhomogeneous diffusion of an RE alloy results in the core-shell structure of grains. On the basis of a thermodynamic assessment, two kinds of core-shell structures were designed for discussion, as shown in Figure 6a–f. The hard-shell grain consisted of an Nd_2_Fe_14_B core and an NdDyFe_14_B shell, and the soft-shell grain consisted of an Nd_1_._64_Dy_0_._36_Fe_14_B core and an Nd_2_Fe_14_B shell. The magnetic parameters of magnetization, exchange constant, and anisotropic constant were calculated on the basis of the Nd and Dy ratios in the τ1 phases. The coercivity of the two models with core-shell grains in Figure 6b was significantly enhanced in comparison with the previous results with homogeneous grains. The magnetization reversals of the models in Figure 6e,f show that the nucleation sites and propagation processes were different due to the modified composition of the core-shell grains. This indicates that the magnetic property distributions of each isolated grain could influence the magnetization reversal in the whole magnet. The available core-shell structures may not be the optimal solution, but it inspired us to improve the microstructure of Nd–Fe–B-based magnets in multiple dimensions. The nucleation sites of core-shell grains were also achieved, and the 2D selected planes from the 3D domains during magnetization reversal are shown in Figure 6c,d. In both situations, the nucleation sites were mainly formed at the center of the grains. This could be understood due to both the pinning effect of nonmagnetic GBs, and the close anisotropic field between core and shell phases.

## 3. Conclusions

In summary, Nd–Dy–Fe–B multicomponent system was systematically investigated with a thermodynamic assessment and micromagnetic simulations. Critical evaluations and optimizations of the available experimental thermodynamic properties and phase diagram data of the Nd–Dy–Fe–B system were performed to build the thermodynamic basement. Thermodynamic analysis with the calculated phase diagrams and transformation-driven force revealed that the partial substitution of Nd with Dy during the preparation of Nd–Fe–B-based magnets is beneficial to the formation of the τ1 phase owing to its inhibitory effect on γ-Fe and RE_2_Fe_17_ phases. In order to achieve a better performance of magnetic properties using a slight amount of Dy or another HRE element, the magnetization reversal behaviors of 3D micromagnetic models were researched by changing the magnetic properties of the GBs and the core-shell grains, which are directly correlative with the distribution of the Dy element in magnets. Nonmagnetic and hard-magnetic GBs could both act as pinning sites during magnetization reversals, and suppress the reversed domain expansion. Coercivity was enhanced in both the hard- and the soft-shell structures. The different magnetization reversal behaviors indicated that the magnetic property distribution in grains is also important for the microstructural design of magnets, which could further influence the demagnetization process and coercivity.

## Figures and Tables

**Figure 1 materials-15-07648-f001:**
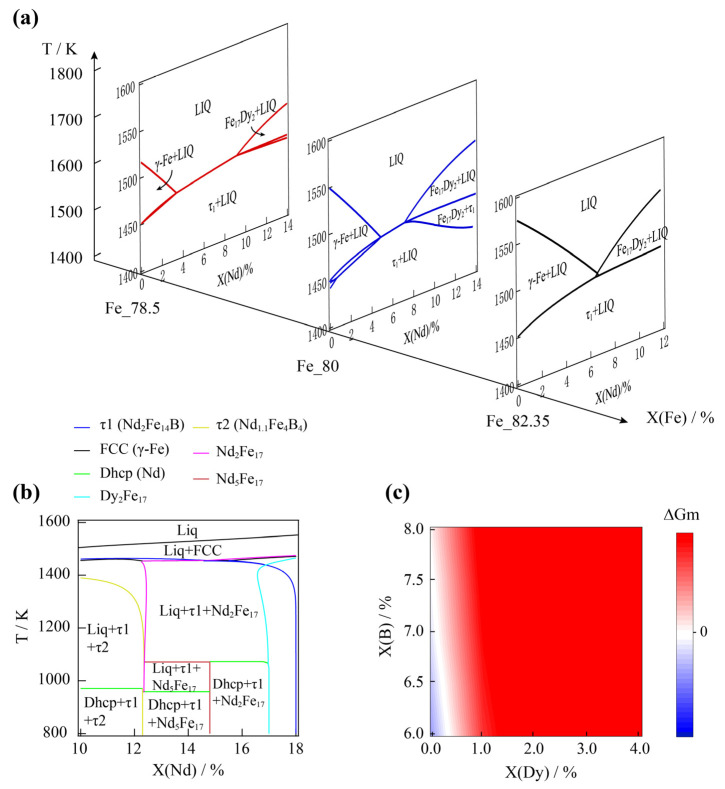
Thermodynamic phase diagrams of the Nd–Dy–Fe–B system. (**a**) Vertical sections of Nd_14_Fe_78_._5_B_7_._5_–Dy_14_Fe_78_._5_B_7_._5_, Nd_14_Fe_80_B_6_–Dy_14_Fe_80_B_6_, and Nd_2_Fe_14_B–Dy_2_Fe_14_B with various amounts of Nd and Dy under temperature in the range of 1400–1800 K. (**b**) Vertical section of Nd–Dy–Fe–B phase diagram in the X(Nd) range of 10–18% with the composition of X(Fe) = 80% and X(Dy) = 2%. (**c**) ∆Gm distribution contour against X(B) and X(Dy); ∆Gm is the difference value of the transformation-driven force between the τ1 and RE_2_Fe_17_ phases.

**Figure 2 materials-15-07648-f002:**
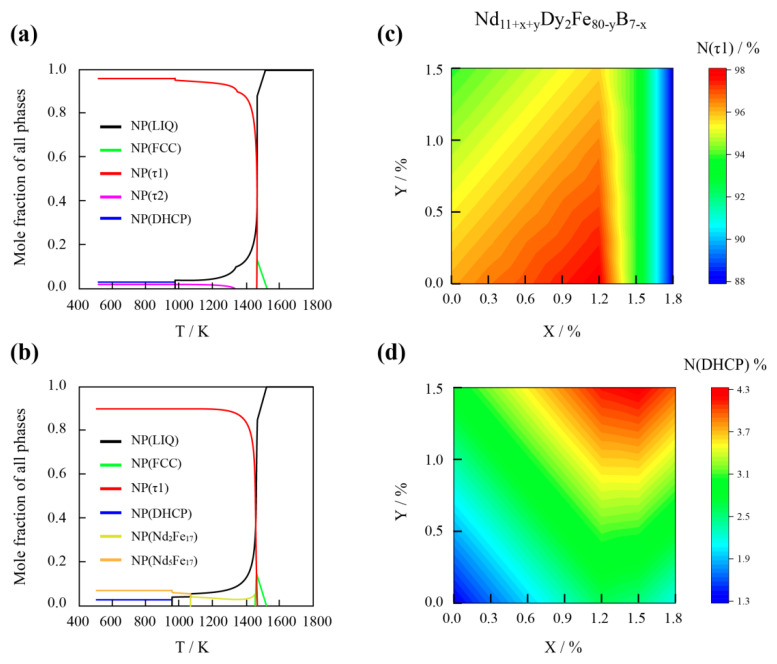
Compositional distributions of τ1 and DHCP phases. Calculated equilibrium continuous cooling diagrams of (**a**) Nd_12_Dy_2_Fe_80_B_6_ and (**b**) Nd_13_Dy_2_Fe_79_._7_B_5_._3_. Mole fraction distributions of (**c**) τ1 and (**d**) DHCP phases against parameters x and y in the Nd_11+x+y_Dy_2_Fe_80–y_B_7–x_ system.

**Figure 3 materials-15-07648-f003:**
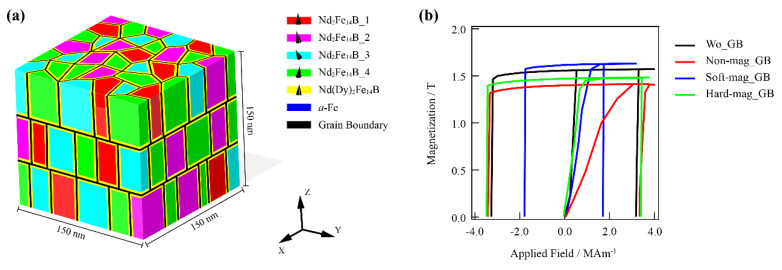
(**a**) Schematic microstructure and magnetic structure of the simulated 3D model used for micromagnetic simulations. Distinct colors indicate specific phases and magnetic orientations. Magnetic orientations of Nd_2_Fe_14_B_1, Nd_2_Fe_14_B_2, Nd_2_Fe_14_B_3, Nd_2_Fe_14_B_4, and Nd(Dy)_2_Fe_14_B were (0.015, 0.085, 0.996), (0.173, 0.015, 0.985), (0.146, 0.253, 0.858), (0.114, −0.042, 0.992), (0.033, 0.012, 0.999), respectively. (**b**) Simulated demagnetization and initial magnetization curves of the models with different types of GBs, as wo_GB, non-mag_GB, soft-mag_GB, and hard-mag_GB.

**Figure 4 materials-15-07648-f004:**
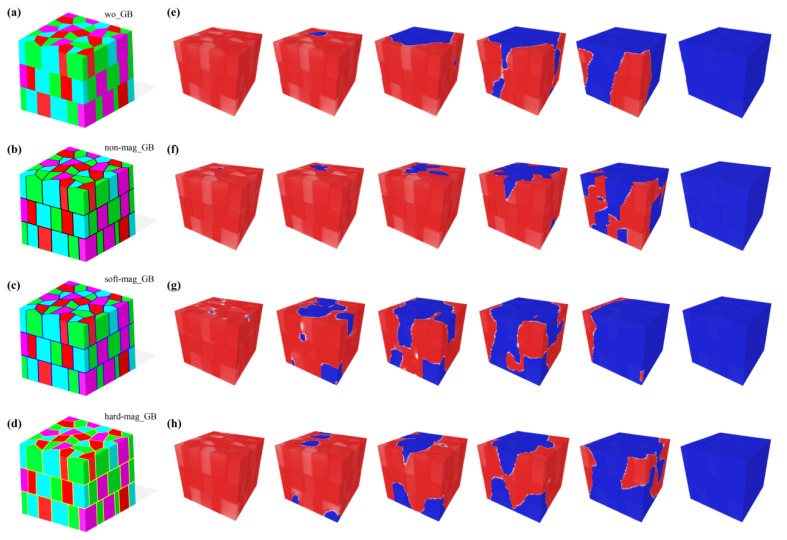
Evolution of 3D domain structures during magnetization reversal. Three-dimensional microstructures of models with different types of GBs: (**a**) without GB, (**b**) nonmagnetic GB, (**c**) soft-magnetic GB, and (**d**) hard-magnetic GB. (**e**–**h**) Three-dimensional domain structures of models corresponding to the microstructures in (**a**–**d**) during the propagation of reversed domains. All these dynamic processes were gathered around the coercivity of each microstructural model.

**Figure 5 materials-15-07648-f005:**
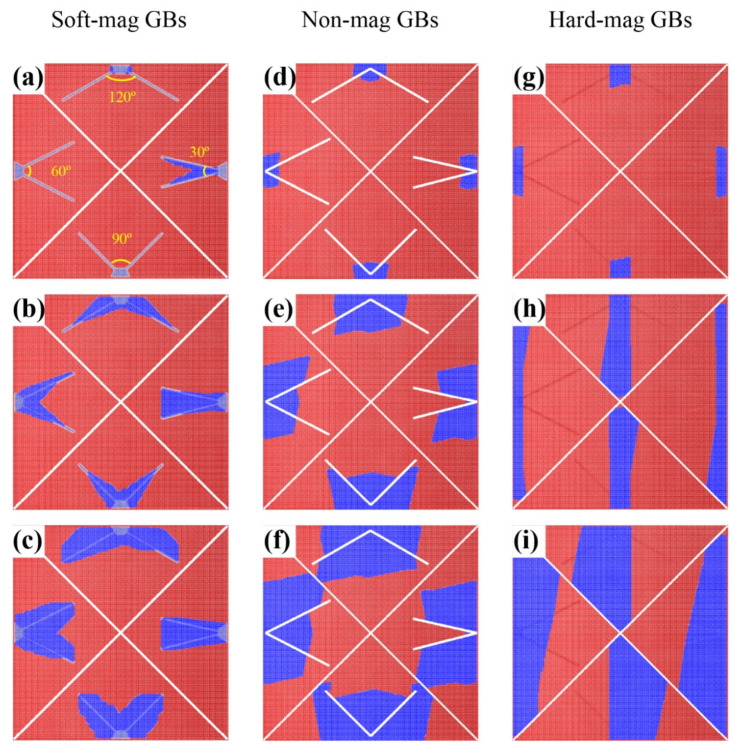
Propagation of reversed domains around 30°, 60°, 90° and 120° GB included angles. (**a**–**c**) Soft-magnetic GBs; (**d**–**f**) nonmagnetic GBs; (**g**–**i**) hard-magnetic GBs. The initial nucleation site was set at the vertex of each angle.

**Figure 6 materials-15-07648-f006:**
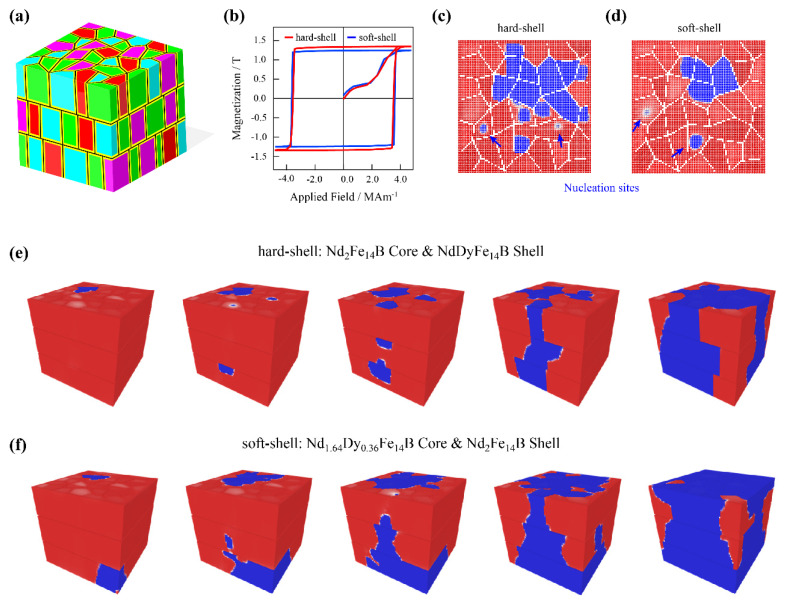
Simulated magnetization reversals of the 3D micromagnetic models with core-shell grains. (**a**) Schematic 3D model consisting of polyhedral core-shell grains. (**b**) Hysteresis loops of models with hard and soft shells. (**c**,**d**) Two-dimensional selected plane capture from 3D domain structures of hard- and soft-shell models during demagnetization. (**e**,**f**) Three-dimensional domain structures of hard- and soft-shell models during the propagation of reversed domains. Hard- and soft-shell grains consisting of an Nd_2_Fe_14_B core with an NdDyFe_14_B shell, and an Nd_1_._64_Dy_0_._36_Fe_14_B core with an Nd_2_Fe_14_B shell.

**Table 1 materials-15-07648-t001:** Optimized model parameters for the Nd–Dy–Fe–B system. *L* denotes the interaction parameters for the excess molar Gibbs energy, and *G* denotes the molar Gibbs energy.

Phases	Thermodynamic Parameters
Liquid	L0Dy,NdLiq=700
	L0B,Dy,FeLiq=−75,000 L1B,Dy,FeLiq=−900
	L2B,Dy,FeLiq=−178,000
BCC	L0Dy,NdBCC=680
HCP	L0Dy,Ndhcp=−1550+0.61T (298K ≤ T ≤ 3000K)
DyNd	G0Dy,NdDyNd=−532.7−10.57T+G0Dyhcp+G0NdDhcp (298K≤ T ≤ 3000K)
Dy_2_Fe_14_B	G0Dy,NdDyNd=−28,969+11.3T+0.117648G0Dyhcp+0.823528G0Fefcc+0.058824G0Brho (800K≤T≤3000K)
Dy_1_._1_Fe_4_B_4_	G0Dy,NdDyNd=−50,410+11T+0.120844G0Dyhcp+0.439078G0Fefcc+0.439078G0Brho (800K≤T≤3000K)

**Table 2 materials-15-07648-t002:** Material parameters of each phase for micromagnetic simulations. α-Fe is the cubic structure, and K1 is a typical anisotropy constant of the soft-magnetic phase [15].

Component	A/pJ/m	K_1_/MJ/m^3^	μ_0_M_s_/T
Nd_2_Fe_14_B	7.7	4.5	1.6
Dy_2_Fe_14_B	6.3	4.0	0.71
Nd-rich	4	0	0
α-Fe	25	0.046	2.1

## Data Availability

Not applicable.

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
