# Peer review of "Magnetic-Property Assessment on Dy–Nd–Fe–B Permanent Magnet by Thermodynamic Calculation and Micromagnetic Simulation"

_materials, 2022, doi:10.3390/ma15217648_

Round 1

Reviewer 1 Report

The authors report  the theoretical calculations of magnetic properties  in Dy-Nd-Fe-B  permanent magnets by thermodynamic and micromagnetic  simulations. The paper is interesting and may be useful in contemporary permanent magnets design. The paper is worth publishing in Materials, but the authors should consider some drawbacks I noticed:

·         Spelling mistake in the line 61; powders probably instead of powers,

·         Thermodynamic parameter L in Table 1 is not defined in the manuscript.

Author Response

Thank you very much for the reviewers’ suggestions on our paper (materials-1959047).  The following point by point responses have been made:

The authors report the theoretical calculations of magnetic properties in Dy-Nd-Fe-B permanent magnets by thermodynamic and micromagnetic simulations. The paper is interesting and may be useful in contemporary permanent magnets design. The paper is worth publishing in Materials, but the authors should consider some drawbacks I noticed:

Spelling mistake in the line 61; powders probably instead of powers,

Reply: Based on the reviewer’s advice, the spelling mistake of ‘powders’ has been corrected in the line 11 of page 3.

Thermodynamic parameter L in Table 1 is not defined in the manuscript.

Reply: Thanks for the reviewer’s advice, the supplemented sentence “L denotes interaction parameters for the excess molar Gibbs energy and G denotes the molar Gibbs energy.” has been added in Table 1 of page 15.

The paper has considerably improved after the extensive revision which has been carried out in full accordance with all reviewers’ comments.

Reviewer 2 Report

The authors present an interesting article about a thermodynamics model and a micromagnetic model. The article can be accepted as long as several improvements are done to the manuscript to increase the interest for the reader.

Some remarks

-Line 37 the units of Ha is A/m. The authors refer to mu0Ha.

-There is a lack of literature about the effect of boundary type.

I think the authors can refer to the following article and some of the references included therein.

https://journals.aps.org/prmaterials/pdf/10.1103/PhysRevMaterials.3.084406

-Page 2, lines 57-59 ”Since the presence of defects or roughness at the τ1-

phase surfaces of the sintered magnets is assumed to be the nucleation site for the de-

magnetization and therefore decrease the coercivity, it can be repaired by the addition of

HRE elements.” “Since” means that one implies the other. I suggest the authors split into two sentences. Something like: “The presence of defects or roughness at the τ1-

phase surfaces of the sintered magnets is assumed to be the nucleation site for the de-

magnetization and therefore decrease the coercivity. Due to this, it can be repaired by the addition of HRE elements.” This statement requires a citation.

–Page 2 Line 65-67 ”Since the phase constitutions and morphology of the microstructures intimately depend on the chemical compositions of the magnets and the different heat treatment processes.” This sentence is incomplete and again it is potential for a citation.

-Page 4. I recommend the authors to increase all the fonts in Fig.1(a). I can scarcely read the labels

-Page 5 line 147. “when the metastable cooling temperature is set below 1200 K” What the authors mean by this?

-Page 5 line 150. “Taken into account” Do the authors mean “included”?

-Page 5 line 172. If the authors used oommf it is better that they mention that explicitly in the main text. Otherwise they have to explain how they integrated the LLG equation.

-Page 6 Line 176. The authors have used a mixture of the energy and field representation but it is not consistent. The part of the anisotropy and exchange energy should not contain a ½ factor. The term of anisotropy consists of a ^2 factor. Please fix it according to https://math.nist.gov/~MDonahue/talks/mcsd20030916-exchange-a4.pdf

There the term of anisotropy is positive because it follows a different convention. In your equations are negative as they should be.

-Page 6 table 2. The authors used Ku1 for alpha-Fe. However,  alpha-Fe is cubic so it has cubic anisotropy. It is fine but then they have to mention that they have used the anisotropic interpolation of the cubic anisotropy as mentioned in Skomski books.

-The authors do not describe what type or orientation u (the anisotropy in ) in the grains have. It looks like they try to indicate this but this is not at all clear for me. I suggest a detailed explanation of the orientation of u.

-Page 7 Figure 4. It would be useful to include the field values for each line of (e-h). I suppose this corresponds to the coercive field but it is better to mention that in either the caption or the figure itself.

-Page 7 line 210. “saturated field” I think the authors mean “saturating field”.

-Page 7 line 215. “with a large corner.” Do the authors mean with “a large angle”?

-Page 7 line 219. “The propagation process indicates that the magnetization reversals are dominated by both the external field and demagnetization field.” Why do the authors conclude that? If that would be the case, I would expect a less bistable switching with a reversible part. Moreover, the authors claim later that uncoupled grains behave like SW particles. These sentences are confusing to me and I don’t know how they reach such a conclusion.

-Page 7 line 221. The authors mention the typical size of the grain boundary but they did not include the size of grain boundaries used in the simulation.

-Page 8 line 226. “Et al.” normally for such expressions is better “etc.”

-Page 8 figure 5. The authors mention 30,60, 90 and 120. But in the figure there are only 3 panels. Also I recommend to include the angle in a column close to the figure as it is done for the type of boundary.

-Page 9 In the discussion of the difference between the different cases it is better to include some numbers. Otherwise, it is difficult to quantify, it looks like everything happens around 40 kOe but how big the difference is is difficult to appreciate.

-All the graphs: The author used for applied field kOe and T in the figures of the loops. This mixture of CGS and SI units is not necessary because they used mostly SI units. I recommend using applied fields in T (Bfield) or A/m (Hfield).

Author Response

The paper has considerably improved after the extensive revision which has been carried out in full accordance with all reviewers’ comments.

Reviewer 3 Report

Manuscript ID: materials-1959047

Reviewer comments.

Dai et al., have conducted a study entitled, “Magnetic properties assessment on Dy-Nd-Fe-B permanent magnet by thermodynamic calculation and micromagnetic simulation”. The authors performed a theoretical study on the phase composition, microstructure, magnetic properties, and coercive mechanisms of the Nd-Dy-Fe-B multicomponent system using the calculation of the phase diagram method and micromagnetic simulation. The partial replacement of Nd by Dy during the synthesis of the Nd-Fe-B based magnets is advantageous to the formation of the τ1 phase due to the inhibitory influence on -Fe and RE2Fe17 phases, according to thermodynamic analysis using the predicted phase diagrams and transformation-driven force.

I found the manuscript scientifically spotless. I would highly recommend this manuscript to be published in MDPI Materials which is a hard-core journal in the field of physics, materials science, and semiconductors with following minor suggestions:

1-      line # 4, the affiliation marking "1,2"etc must be in superscript

2-      line # 19 & 20, put the numbers in superscript

3-      line # 60 & 61, is it "power" or "powder"?

4-      Figure. 1 resolution and text of scales and labelling should be increased for clear visibility

Author Response

Thank you very much for the reviewers’ suggestions on our paper (materials-1959047). According to the comments of the referees, the following point by point responses have been made:

1- line # 4, the affiliation marking "1,2"etc must be in superscript

Reply: The affiliation marks has been marked in superscript. In word document, these marks are in superscript.

2- line # 19 & 20, put the numbers in superscript

Reply: At line 19 and 20, the Nd2Fe17 and Nd2Fe14B have been revised and the format is corrected in word document.

3- line # 60 & 61, is it "power" or "powder"?

Reply: Based on the reviewer’s advice, the spelling mistake of ‘powders’ has been revised in the line 11 of page 3.

4- Figure. 1 resolution and text of scales and labelling should be increased for clear visibility

Reply: The figure 1 has been revised and the fonts are increased.

The paper has considerably improved after the extensive revision which has been carried out in full accordance with all reviewers’ comments.

Reviewer 4 Report

Comments to the Author: The manuscript is well written, the results are well presented and discussed upon. Nevertheless, some aspects need to be further discussed in order for the manuscript to be suitable for publication.
1- The Abstract is somewhat dry; it should include more actual results from the manuscript.
2- The Authors should also consider rephrasing some of the paragraphs, to improve the manuscript in terms of the English language.
3-
Could the authors compare their results with the properties of the other materials?

4. Here some references talking about the preparation of such these materials

·         Tailoring optical, magnetic and electric behavior of lanthanum strontium manganite La 1− x Sr x MnO 3 (LSM) nanopowders prepared via a co-precipitation method with different Sr 2+ ion contents , Rsc Advances 6 (22), 17980-17986

·         Optical, electrical and magnetic properties of lanthanum strontium manganite La 1− x Sr x MnO 3 synthesized through the citrate combustion method, Physical Chemistry Chemical Physics 19 (9), 6878-6886

A Robust and Highly Precise Alternative against the Proliferation of Intestinal Carcinoma and Human Hepatocellular Carcinoma Cells Based on Lanthanum Strontium Manganite, Materials 14 (17), 4979
For the above comments I will recommend this manuscript as major revision

Author Response

1- The Abstract is somewhat dry; it should include more actual results from the manuscript.

Reply: Based on the reviewer’s advice, the actual results have been added. The sentence “It is found that γ-Fe and Nd2Fe17 phases will be suppressed when partial Nd is substituted by Dy, which in turn makes the formation of Nd2Fe14B phase become favorable.” has been changed into “It is found that γ-Fe and Nd2Fe17 phases will be suppressed when partial Nd is substituted by Dy (< 9.3%), which in turn makes the formation of Nd2Fe14B phase become favorable.” and the sentence “The coercivity enhancement is obtained with both the non-magnetic and hard-magnetic GBs for the pinning effect besides the GBs.” has been changed into “The coercivity enhancement is obtained with both the 3 nm non-magnetic and hard-magnetic GBs for the pinning effect besides the GBs.” in the Abstract.

2- The Authors should also consider rephrasing some of the paragraphs, to improve the manuscript in terms of the English language.

Reply: The manuscript has been revised again, and changed paragraphs were signed with blue color. I think the quality of the language has been improved.

3- Could the authors compare their results with the properties of the other materials?

Reply: The thermodynamic data has been compared with the results of Grieb, Kobzenko, and Wang, et al., which has been shown in the supplied information. For the micromagnetic simulations, all the parameters have been proved to be right by a lot of theoretical and experimental works, and the simulated results have been compared with the experimental results of Sepehri-Amin and Hono, et al., which are highly agree with the experimental regular.

  1. Here some references talking about the preparation of such these materials

Tailoring optical, magnetic and electric behavior of lanthanum strontium manganite La 1− x Sr x MnO 3 (LSM) nanopowders prepared via a co-precipitation method with different Sr 2+ ion contents , Rsc Advances 6 (22), 17980-17986

Optical, electrical and magnetic properties of lanthanum strontium manganite La 1− x Sr x MnO 3 synthesized through the citrate combustion method, Physical Chemistry Chemical Physics 19 (9), 6878-6886

A Robust and Highly Precise Alternative against the Proliferation of Intestinal Carcinoma and Human Hepatocellular Carcinoma Cells Based on Lanthanum Strontium Manganite, Materials 14 (17), 4979

Reply: Based on the reviewer’s advice, the reference “[11] Turky, A.O.; Rashad, M.M.; Hassan, A.M.; Elnaggar, E.M.; Bechelany, M. Tailoring optical, magnetic and electric behavior of lanthanum strontium manganite La1−xSrxMnO3 (LSM) nanopowders prepared via a co-precipitation method with different Sr2+ ion contents. RSC Adv. 2016, 6, 17980-17986.; [12] Turky, A.O.; Rashad, M.M.; Hassan, A.M.; Elnaggar, E.M.; Bechelany, M. Optical, electrical and magnetic properties of lanthanum strontium manganite La1−xSrxMnO3 synthesized through the citrate combustion method. Phys. Chem. Chem. Phys. 2017, 19, 6878-6886.; [13] Turkey, A.O.; Abdelmoaz, M.A.; Hessien, M.M.; Hassan, A.M.; Bechelany, M.; Ewais, E.M.; Rashad, M.M. A robust and highly precise alternative against the proliferation of intestinal carcinoma and human hepatocellular carcinoma cells based on lanthanum strontium manganite nanoparticles. Materials 2021, 14, 4979.” has been added in this manuscript.